# A Framework for Assessing the Impact of Outbreak Response Immunization Programs

**DOI:** 10.3390/diseases12040073

**Published:** 2024-04-04

**Authors:** Dominic Delport, Ben Sanderson, Rachel Sacks-Davis, Stefanie Vaccher, Milena Dalton, Rowan Martin-Hughes, Tewodaj Mengistu, Dan Hogan, Romesh Abeysuriya, Nick Scott

**Affiliations:** 1Burnet Institute, Melbourne, VIC 3004, Australia; ben.sanderson@burnet.edu.au (B.S.); rachel.sacks-davis@burnet.edu.au (R.S.-D.); stefanie.vaccher@burnet.edu.au (S.V.); milena.dalton@burnet.edu.au (M.D.); rowan.martin-hughes@burnet.edu.au (R.M.-H.); romesh.abeysuriya@burnet.edu.au (R.A.); nick.scott@burnet.edu.au (N.S.); 2Department of Epidemiology and Preventive Medicine, Monash University, Melbourne, VIC 3004, Australia; 3School of Population and Global Health, The University of Melbourne, Parkville, VIC 3010, Australia; 4Gavi, The Vaccine Alliance, 1218 Geneva, Switzerland; tmengistu@gavi.org (T.M.); dhogan@gavi.org (D.H.)

**Keywords:** infectious diseases, vaccine preventable diseases, outbreak response immunization, measles, Ebola, modelling, full value of vaccine assessment, impact, value, economic

## Abstract

The impact of outbreak response immunization (ORI) can be estimated by comparing observed outcomes to modelled counterfactual scenarios without ORI, but the most appropriate metrics depend on stakeholder needs and data availability. This study developed a framework for using mathematical models to assess the impact of ORI for vaccine-preventable diseases. Framework development involved (1) the assessment of impact metrics based on stakeholder interviews and literature reviews determining data availability and capacity to capture as model outcomes; (2) mapping investment in ORI elements to model parameters to define scenarios; (3) developing a system for engaging stakeholders and formulating model questions, performing analyses, and interpreting results; and (4) example applications for different settings and pathogens. The metrics identified as most useful were health impacts, economic impacts, and the risk of severe outbreaks. Scenario categories included investment in the response scale, response speed, and vaccine targeting. The framework defines four phases: (1) problem framing and data sourcing (identification of stakeholder needs, metrics, and scenarios); (2) model choice; (3) model implementation; and (4) interpretation and communication. The use of the framework is demonstrated by application to two outbreaks, measles in Papua New Guinea and Ebola in the Democratic Republic of the Congo. The framework is a systematic way to engage with stakeholders and ensure that an analysis is fit for purpose, makes the best use of available data, and uses suitable modelling methodology.

## 1. Introduction

There have been significant gains in immunisation coverage in recent decades; however, outbreaks of vaccine-preventable diseases (VPDs) still have devastating health, economic, and social impacts, particularly in low- and middle-income countries (LMICs) [1,2]. There were over 14 million children in 2022 who did not receive a vaccine for any disease, and 1.5 million people lose their lives to VPDs each year [3,4]. For example, the 2014–2016 Ebola virus outbreak in West Africa led to more than 28,600 cases and 11,310 deaths across Guinea, Sierra Leone, and Liberia [5] and is estimated to have caused USD 2.8 billion in economic damages [6]. Africa, in particular, is the region with the highest numbers of unvaccinated children, with 7.7 million children aged 12 to 23 months not having received any vaccines in 2022 [7], which is more than 50% of the burden of unvaccinated children globally [3]. Different interventions exist to either minimize outbreak occurrence (e.g., preventative vaccination, health system strengthening) or to minimize the scale of outbreaks when they do happen (e.g., vaccine stockpiles to ensure a timely response and containment measures such as outbreak response immunization (ORI) programs) [2,8]. The Sustainable Development Goals targets 3.2 and 3.3 call for significant reductions in neonatal and child deaths and the combatting of communicable diseases [9]. Interventions which can effectively prevent or minimize the scale of disease outbreaks, particularly those where the burden is focused on children, can be valuable tools for reaching these targets.

The impact of outbreaks of VPDs varies substantially, ranging from a small number of cases to uncontrolled spread, causing social and economic disruption [10,11]. Compared to evaluating the impact of preventive immunization programs, the unpredictable frequency and stochastic nature of outbreaks make it difficult to assess the impact of investment in vaccine stockpiles and the ORI programs which use them [12]. Containment measures, including the distribution of stockpiled vaccines, influence the scale of the outbreak [13], and upfront investment is required to ensure sufficient stockpiles are in place. However, they may never be needed (i.e., when there is no outbreak), or when they are needed, it is not clear what scale of outbreak would have happened in their absence.

Mathematical models have been used for public health applications for many years to analyse disease outbreaks and responses to them [1,10,14,15,16,17,18]. Historically, models of influenza, measles, malaria, and many other diseases have been used to inform outbreak response and health policy [1,10,17,18], and more recently, responses to the COVID-19 pandemic often made use of estimates from modelling groups [10,19]. When considering a historical outbreak, a model ‘baseline’ scenario can be calibrated to empirical data and then compared to counterfactual scenarios that estimate possible outcomes had different responses occurred [1]. By varying elements of response strategies (e.g., vaccination), counterfactual scenarios can help quantify how each strategy contributed to ending the outbreak. This provides an evidence base to aid in the selection of effective strategies for future outbreaks [20].

When comparing observed outcomes of outbreaks to modelled counterfactual scenarios, it is not clear which metrics best capture the impact of an ORI program, which outcomes are useful for the organisations involved in funding or coordinating the program, and what data limitations mean for the feasibility of modelling these outcomes. To help policymakers and modellers address these issues systematically, we developed a framework for using mathematical models to assess the impact of ORI programs and produce outputs to inform future investments, advocacy, and resource mobilization. This conceptual framework captures and explains the key factors involved in the analysis process and links them together [21,22,23,24]. Previously developed frameworks in the health economics space identify the importance of communication with stakeholders as a part of the modelling process [25,26]. We propose to formalize this involvement of stakeholders in the structure of the framework.

In the Methods section, we describe the approaches used to inform and design our new framework. In the Results section, we present the framework and demonstrate its usage through application to two historical outbreaks.

## 2. Materials and Methods

### 2.1. Overview of Framework Development

The framework was developed by (1) conducting semi-structured interviews with selected individuals from global health organisations to identify which model outputs they considered most useful; (2) undertaking literature reviews to scope the availability of data for historical outbreaks and feasibility of estimating these outputs; and (3) using these results to inform a selection of impact summary metrics, proposed model scenarios, and process for stakeholder engagement alongside technical decisions related to the modelling. The framework was designed to ensure scenarios and model choices are selected based on available data and that the expected outputs meet the needs of stakeholders and can be communicated clearly and impactfully. Two modelling studies were then performed as illustrative examples.

### 2.2. Semi-Structured Interviews

Individuals from global health organizations were selected and invited to participate, such that participants collectively included a broad range of experience (clinical, epidemiology, public health, policy) and functions within their respective organizations (e.g., program implementation, resource mobilization, monitoring, and evaluation). Participants were from a variety of organizations, including Gavi, the Vaccine Alliance (Gavi), the University of Cambridge, the United States Centers for Disease Control and Prevention, the World Bank, the Bill and Melinda Gates Foundation, the World Health Organization, United Nations Children’s Fund, and the International Organization for Migration. Semi-structured interviews were conducted with individuals or small groups (2–5 people) from the same organization. In the interviews with a small group from the same organization, a single overall response was recorded.

Before the interviews a set of seven measures were identified by the authors based on experience with investment cases. Interviews were semi-structured, and in each interview, facilitators prompted participants to reflect on the strengths and weaknesses of each measure and score them as either “not at all useful”, “slightly useful”, “somewhat useful”, “very useful”, or “extremely useful” (Appendix A). Participants were also asked for suggestions of additional measures which they considered useful. Distributions of survey responses were produced, and a thematic analysis was undertaken on participants’ reflections and explanations during interviews.

### 2.3. Ethics

Ethics approval for this study was obtained from the Alfred Hospital Melbourne (project 636/22). Participant consent was given verbally before interviews were started, all participants were advised of their right to withdraw from the study at any time, and no identifiable information was used.

### 2.4. Data Availability Review

A literature review was undertaken using a systematic approach (methods detailed in Appendix A) to identify how frequently different epidemiological, health system, cost, and program response measures are reported for VPD outbreaks. We identified data that are typically recorded and publicly available following outbreaks in LMICs and would, therefore, be feasible to collect when an analysis is being conducted. To capture data available from the grey literature, such as government reports, as well as the academic literature, searches were conducted using Google and Google Scholar.

The search was performed in three parts: first, the identification of the frequency with which different epidemiological measures are reported; second, a search for measures of health system impacts and costs incurred during outbreaks; and third, a search for ORI program data. The searched VPDs focus primarily on diseases with frequent outbreaks (such as measles), diseases for which Gavi supports ORI or vaccine stockpiles (measles, Ebola, yellow fever, meningitis, and cholera), and ‘other’ VPDs (i.e., mumps, pertussis, polio, hepatitis A, diphtheria). Appendix A provide the collections of search terms for each part.

### 2.5. Scenarios Defined for Use within the Framework

When performing a modelling analysis, scenarios are defined by changes in one or more model parameters. For the development of the framework, components of ORI programs that could be invested in were identified (e.g., stockpile size) and then grouped according to which model parameters they would influence (e.g., ORI coverage).

### 2.6. Example Analyses

Two analyses were performed to illustrate how the framework can be applied to evaluate ORI programs in LMICs. The analyses were for an outbreak of measles in Madang province of Papua New Guinea (PNG) in 2014–2015 and an outbreak of Ebola in Equateur province of the Democratic Republic of the Congo (DRC) in 2020. These examples were selected to demonstrate how the framework is applicable across settings and pathogens, as well as different types of models (a stochastic compartmental model for measles and an agent-based model for Ebola).

## 3. Results

### 3.1. Key Impact Metrics

A series of 16 qualitative interviews were conducted over Zoom throughout November and December 2022 with a total of 25 participants.

Participants considered metrics capturing epidemiological and health outcomes such as cases and deaths averted the most useful outputs for informing advocacy and the impacts of future investment, with more than 90% of respondents considering them to be “very useful” or “extremely useful” (Figure 1). Measures of economic impact and outbreak risk, such as the cost of illness averted or the probability that outbreaks can be ended early, were also considered to have high utility by most interviewees, but with varied responses depending on participants’ backgrounds.

The selected metrics were split into three categories: health impacts (cases, deaths, and disability-adjusted life years (DALYs) averted), economic impacts (direct healthcare costs and indirect economic costs [e.g., societal cost of years of life lost]), and risk and disruption (probability of a severe outbreak [i.e., outbreaks exceeding different case thresholds]). The chosen metrics were those that met the needs of most participants while also having sufficiently available data to inform the model characteristics necessary to output them (Table 1).

Further details of findings from the interviews can be found in Appendix A.

The main themes identified from the interviews were that when considering which outputs were the most useful: simplicity is key, data availability and quality are critical, interviewees’ backgrounds influenced which metrics they considered most valuable, any estimate of health system impact would be useful, and it is important to consider how comparable measures would be across settings and diseases.

### 3.2. Data Availability

Across all VPDs, cumulative deaths and cases were the most reported measures, appearing in 75% and 70% of sources, respectively (all sources included are available in the Appendix A). Where weekly outbreak reports were available, the cumulative cases and deaths could be separated into weekly values, but such frequent reporting was not always the case. Other epidemiological metrics reported included the percentage of samples which were positive and additional disaggregation of cases and deaths (e.g., within healthcare workers, by confirmed or probable, or by district); however, reporting varied across pathogens and outbreaks.

A large body of studies exist which investigate the impacts of Ebola on health systems [27,28]; however, the reported metrics are highly varied and reported with low frequency, with the most reported measure being the change in visits to health care facilities during an outbreak. We were unable to find health system impact data for set of searched VPDs other than Ebola, which indicates that it may be difficult to estimate the impacts of other pathogens.

Overall, different types of direct and indirect economic impacts were frequently reported, appearing in 45% and 35% of recorded sources, respectively. Most of the returned sources during this search were for Ebola, including an analysis by Bartsch et al. [29], describing a methodology for estimating the cost incurred by an Ebola case, capturing both direct health costs incurred by cases and losses to productivity from absenteeism. This could be generalized as an ingredient-based approach to cost appraising other diseases where the health impacts and treatment procedures are sufficiently well understood.

The frequency with which ORI program data were found within the academic and grey literature is lower than the epidemiological, health system, and economic impact data. The most frequently reported measures were the total number of vaccines delivered and the time taken for a vaccine administration response to start after an outbreak is detected, appearing in 76% and 21% of the sources, respectively. These two parameters were expected to be necessary for model calibration to historical outbreaks. Gavi also reports their disbursed funding [30], which are not specific to individual outbreaks, but could help inform programmatic spending if mapped to vaccine allocations from the International Coordinating Group (ICG) on Vaccine Provision [31], and/or unit cost estimates.

Further details including summaries of identified disease, health system, economic impacts, and response measures, disaggregated by disease, are in Appendix A presenting the frequencies with which each measure appeared.

### 3.3. Scenario Categories

Three scenario categories were defined based on the parameters that would vary when investment in different elements of an ORI program occurred:Scale of response. This could improve due to investment in increased supply or stockpiles, decreased wastage, increased workforce training for delivery, the development of an outbreak response plan, or supply-chain readiness.Speed of response. This could improve due to investment in increased supply chain or workforce readiness, improvements to the cold chain, the development of an outbreak response plan, or increased stockpiling (and, hence, no delays on procurement).Prioritization of delivery. This refers to improving the targeting of ORI programs among vulnerable groups or contacts of known cases. This could be achieved by investment in outreach programs or contact tracing capacity the or development of an outbreak response plan.

Investment in elements of an ORI program (e.g., vaccine stockpile size or the time taken to detect an outbreak) can be mapped to model parameters (Figure 2). These parameters (e.g., baseline vaccine coverage or vaccine rollout rate) can be varied to define different scenarios and explore the relative impact on outbreak size, and therefore health, economic, and risk outcomes. The relationship between investment in ORI components and model parameters would depend on the specific context being considered.

Further details for the proposed scenario categories are in Appendix A.

### 3.4. Framework Structure

Synthesising the prior elements of this study, the framework (represented diagrammatically in Figure 3) incorporates scenarios and model choice based on available data, and the selection of outputs that meet the needs of stakeholders and can be communicated clearly and impactfully. The framework articulates how past or potential future investment in ORI programs can be evaluated, beginning from the choice of setting and pathogen and ending with the interpretation and communication of the model outputs.

The framework has four phases: (1) problem framing and data sourcing; (2) model choice; (3) model implementation, including simulating outbreaks and comparing outputs between scenarios; and (4) interpretation and communication (Figure 3). Each phase contains key decisions or tasks, which together are required to conduct an investment case for an ORI program (detailed in Table 2). The ongoing review of evidence; expert advice; and the regular validation of data, parameters, and the model itself are required across all phases. Modelling analyses should consider multiple types of model validation where feasible, such as face (does the model align with current evidence), internal (does the model behave as expected), cross (how does the model compare to similar models), external (do modelled scenarios align with reality), and predictive (are model forecasts accurate) [32].

Further detail on the phases and example considerations for each decision and task are in Appendix A.

### 3.5. Example Analyses

Figure 4 and Figure 5 summarize representative analyses performed using data from the 2014–2015 measles outbreak in Madang Province, PNG [34], and the 2020 Ebola outbreak in Equateur Province, DRC [35]. The four sections in each summary figure align with the phases of the framework described in Figure 3 and Table 2. The figures briefly describe the context of the outbreak and aims of the analysis, the data available to inform the model and a representation of its structure, the scenarios examined and how well the model fits the available data, and key results from the analysis and their interpretation. A detailed description of each analysis can be found in Appendix A, including descriptions of the models used. Appendix A present the parameters used in the measles model and a schematic representation of its structure, while Appendix A do the same for the Ebola model.

The Madang measles outbreak resulted in 5073 cases and 30 deaths. This analysis estimated that the 71,474 vaccines administered by the ORI program led to 35%, 33%, and 32% fewer measles cases, deaths, and DALYs compared to a scenario without ORI. There were 2667 (2432–2900) cases, 14 (12–16) deaths, 402 (344–460) DALYs, and USD 2,238,008 (USD 1,931,824–USD 2,535,783) in health and economic costs prevented, a 4.8-fold return-on-investment. Additional scenario analyses found that underlying vaccine coverage had a significant influence on outcomes; scenarios with baseline (pre-outbreak) vaccine coverage increased by 10% or 20% resulted in 42% or 66% fewer DALYs, respectively. Further details of the calibration, modelled scenarios, and impact estimates can be seen in Appendix A.

The 2020 Equateur Ebola outbreak caused 130 cases and 55 deaths, with the 43,000 vaccines delivered by the ORI program estimated to have reduced Ebola cases, deaths, and DALYs by 19%, 19%, and 18% compared to a scenario without ORI. There were 41 (20–87) cases, 13 (6–27) deaths, 338 (150–711) DALYs, and USD 309,636 (USD 134,534–USD 637,127) in societal economic costs prevented. In addition, the ORI program was estimated to have reduced the risk of the outbreak exceeding 200 cases, corresponding to about USD 1.6 M in the direct health system and indirect societal costs, from 56% to 36%. Further scenario analyses indicated that the targeted delivery of vaccines to known contacts of cases was one of the most impactful features of the program; compared to the non-targeted vaccines scenario, the ring-based vaccination was estimated to have reduced overall cases and deaths by 17% and 16%, respectively, as well as reduced the risk of the outbreak exceeding a threshold of 200 infections by more than 15 percentage points. Further details of the calibration, modelled scenarios, and impact estimates can be seen in Appendix A.

## 4. Discussion

This study developed a framework for using mathematical models to assess the impact of ORI programs for VPDs, providing a standardised approach and set of output metrics. Participants of qualitative interviews identified specific health impacts, economic impacts, and the risk of severe outbreaks as the most useful metrics, with literature reviews identifying them as feasible to estimate. The framework defines four phases: (1) problem framing and data sourcing (identification of stakeholder needs, metrics, and scenarios); (2) model choice; (3) model implementation; and (4) interpretation and communication. Two example use cases, measles in Papua New Guinea (2014–2015) and Ebola in the Democratic Republic of Congo (2020), demonstrated important health and economic benefits associated with ORI programs.

Metrics identified as most useful for evaluating ORI programs could typically be produced by models with commonly available data; however, there were data gaps that could make some metrics, contexts, or diseases more difficult than others. Regular reporting of cases, deaths, and vaccines delivered during outbreaks can be used in models alongside known disease characteristics and setting-specific demographic and epidemiological data to estimate cases, deaths, and societal costs averted by ORI programs. Additional data relating to health system use during outbreaks would enable better estimation of direct health costs, but this was not always identified as a high priority because differences in health system operations and capacity across settings could make it difficult to interpret. Data on broader components of outbreak response programs was not always available either but, if more widely reported, could enable model scenarios to assess the impact and synergies of different elements beyond vaccination.

The framework structure was developed to divide the modelling process into distinct phases covering the analysis process from problem framing and identifying appropriate metrics to result communication. This structure has conceptual similarities to previous work by Squires et al. [25] and Tappenden et al. [26]; however, those frameworks are more technical than we described here, and both come from a modeller-centric perspective. In contrast, a strength of this framework is that it was developed with input from experts with a broader diversity of experience within public health and international development and aims to have a more stakeholder-centric perspective. Interestingly, the formalization of stakeholder input into this framework during development shares parallels with how information dashboards are produced for health systems, indicating that our approach is similar to methods used in other contexts. Several studies describe how the development of dashboards in different medical fields incorporate stakeholder and user feedback so that the final product is user-centred and effective [36,37,38]. Stakeholders are key to translating model findings into impact, and the framework is designed to facilitate producing and communicating metrics which stakeholders consider useful by incorporating their feedback from the beginning of this project. Another strength of the framework is the consideration of risk of large outbreaks averted by ORI programs, which is often overlooked despite being a key benefit.

There are also limitations to the framework. While it was designed to be generalisable and model agnostic, the interpretation and communication of results will always be contextual. Consultation with country teams or experts with knowledge of specific ORI programs is necessary to support any assumptions, particularly when data are unavailable or there is ambiguity as to their interpretation. Depending on the disease characteristics or stakeholder needs, the requirements of an analysis may also expand beyond the methods or output metrics described here. In particular, the data availability search and assessment of outcome metrics was considered primarily in the context of LMICs. However, the findings of this study and the proposed framework should be broadly applicable to outbreaks in high-income countries as well, with potential adjustments required for the input data, output metrics, and interpretation of results for stakeholders. Although the framework incorporates the needs of stakeholders, the model choice or development and analysis stages require varying degrees of technical understanding (depending on the model), which limits its accessibility.

We demonstrated how the framework can be applied within two disease contexts with two different models. From the first example analysis, our results indicated that increasing baseline measles vaccine coverage should remain the highest priority and that early and rapid responses can have the greatest impacts on averting negative outcomes. Conversely, for the second example, the greatest impact on the modelled Ebola outbreak likely came from effective contact tracing and implementation of ring-based vaccination strategies instead of rapid dissemination of vaccines. Program implementors, epidemiologists, and public health experts would likely be most interested in these findings as they indicate where impact can be achieved within the response. However, Ministries of Finance and external funders may be more interested in the estimated 4·8-fold return-on-investment from the measles ORI program, and policymakers interested in the 20-percentage point reduction in the risk of a large outbreak that the Ebola ORI program was estimated to have achieved. The comparison of results from these two applications is useful to illustrate not only that the framework is model agnostic but also how the choice of model can (and should) depend on the pathogen being studied and the types of measures which are to be output. The methods defined by this framework can be generalised and applied to outbreaks of any disease which received outbreak response immunisation.

## 5. Conclusions

Experts identified key health, economic, and risk outcomes associated with outbreaks of VPDs that they valued, with reported data typically available to inform one or more of them; however, data on the extent and cost of public health responses was limited. The framework proposed here provides a structured system for assessing the impact of ORI programs in LMICs, incorporating the impact metrics identified by stakeholders and typical data availability to inform them. The framework can be used by modellers, public health teams, and health systems experts to articulate how investment in ORI programs can be evaluated, beginning from the choice of setting and pathogen, and ending with the interpretation and communication of the model outputs. Its application in two example analyses demonstrated important health and economic benefits associated with ORI programs.

## Figures and Tables

**Figure 1 diseases-12-00073-f001:**
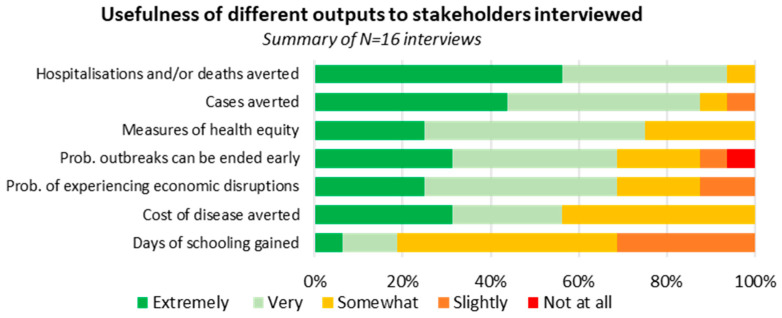
Summary of ranked responses rating utility of output measures proposed for developing an investment case for vaccine stockpiling for outbreak responses.

**Figure 2 diseases-12-00073-f002:**
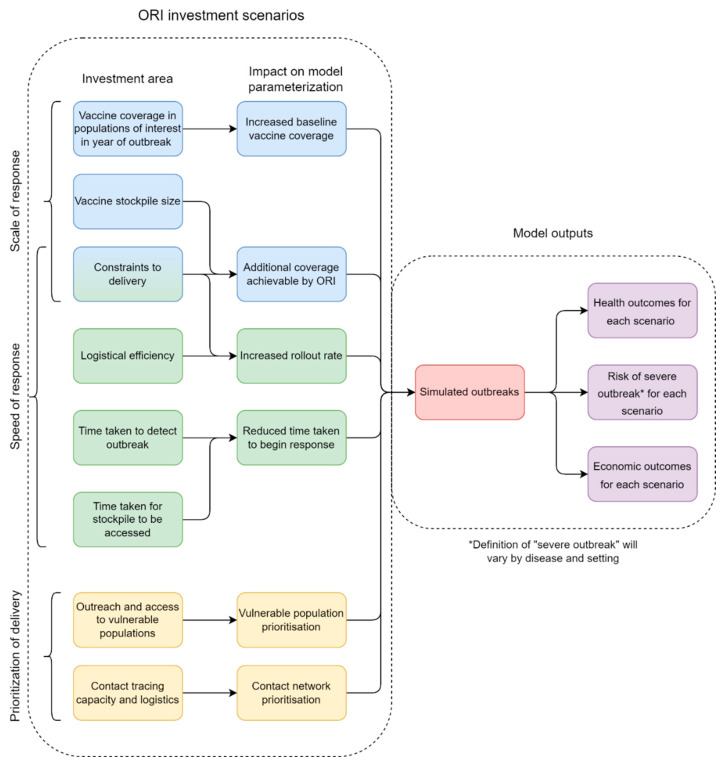
Schematic representation of the proposed types of investment scenarios (Scale of response, Speed of response, and Prioritization of delivery) for outbreak response immunization (ORI) programs to be evaluated by the framework.

**Figure 3 diseases-12-00073-f003:**
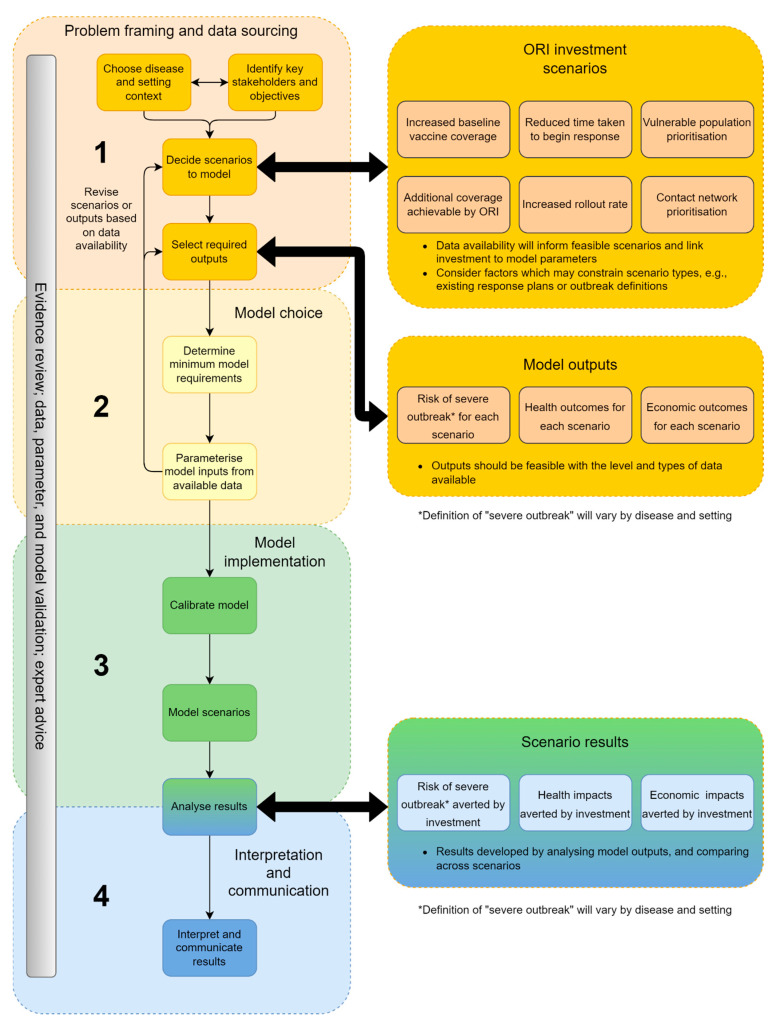
Framework for developing an investment case for ORI programs.

**Figure 4 diseases-12-00073-f004:**
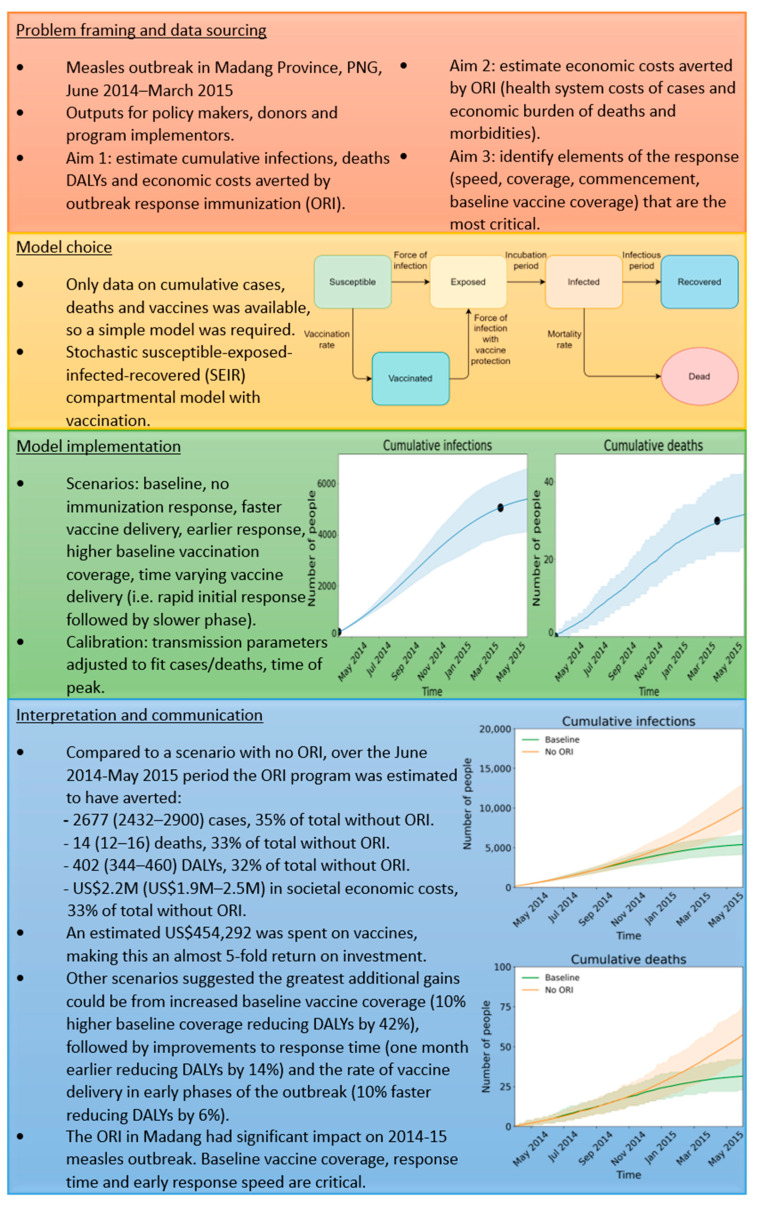
Summary of 2014–2015 Madang Province, PNG measles outbreak analysis. Each coloured section aligns with a phase of the framework and briefly describes details relevant to that phase.

**Figure 5 diseases-12-00073-f005:**
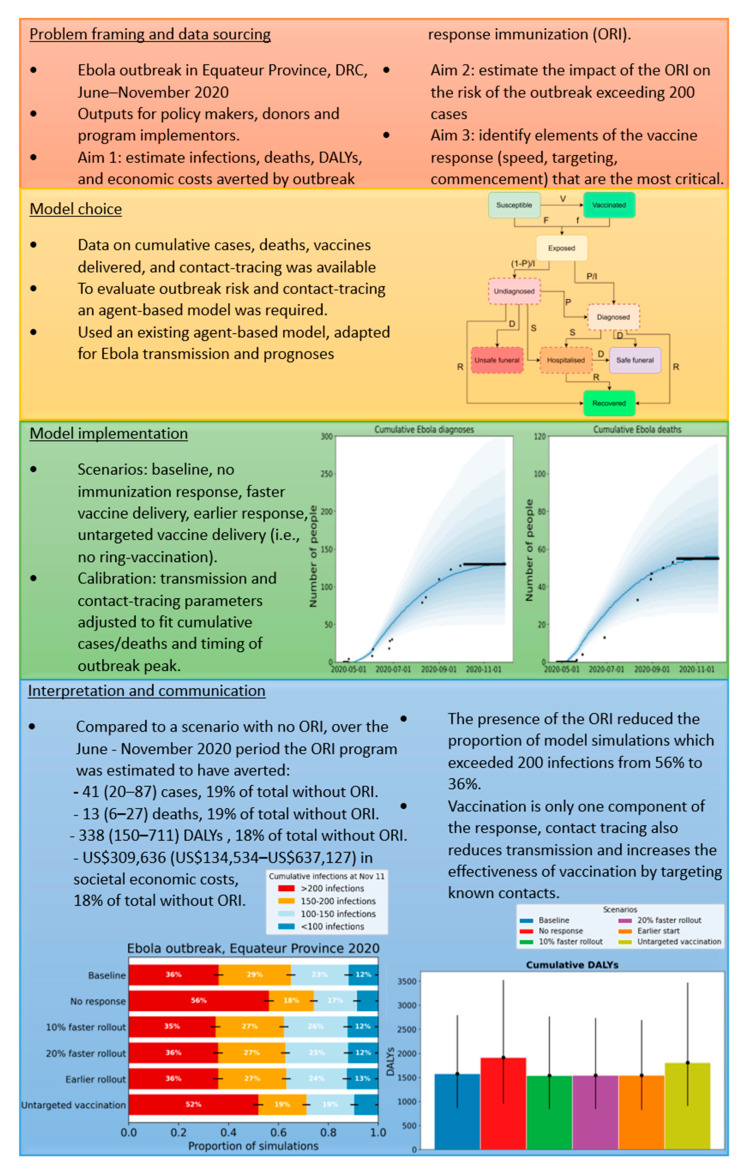
Summary of 2020 Equateur Province, DRC Ebola outbreak analysis. Each coloured section aligns with a phase of the framework and briefly describes details relevant to that phase.

**Table 1 diseases-12-00073-t001:** Summary of stakeholder interest and availability of data for potential measures of a counterfactual outbreak response immunization scenario. Output measures are grouped into four broad categories: health impacts (cases averted, deaths averted, disability-adjusted life years (DALYs) averted, hospitalisations averted), economic impacts (expected healthcare costs averted, expected economic costs averted), risk and disruption (probability of severe outbreak, probability of economic disruption, healthcare system impact, service disruption, impact on neighbouring countries), and other (measures of health equity, days of schooling gained).

Output Measure	Consultation Feedback	Input Data Availability	Proposed Output	Rationale
**Health impacts**				
Cases averted	High interest	Frequently available	Yes	
Deaths averted	High interest	Frequently available	Yes	
DALYs averted *	N/A	Calculated from cases/deaths/disability weights	Yes	Included as they can often be estimated from cases and deaths.
Hospitalizations averted	Moderate interest	Unavailable	No	Can be calculated if sufficient data are available but considered less useful than other health impact measures.
**Economic impacts**				
Expected healthcare costs averted	Moderate interest	Frequently available, and methods exist to estimate	Yes	
Expected economic costs averted	Moderate interest	Regularly available, and methods exist to estimate	Yes	
**Risk and disruption**				
Probability of severe outbreak **	Moderate interest	Theoretical measure; data not applicable	Yes	
Probability of economic disruption	Moderate interest	Theoretical measure; data not applicable	No	These measures are highly context-specific and can be difficult to define. However, for specific use cases, if sufficient relevant information was available they could be estimated from the probability of severe outbreak measure.
Healthcare system impact *	N/A	Some measures frequently available	No	
Service disruption *	N/A	Rarely available	No	
Impact on neighbouring countries *	N/A	Rarely available	No	
**Other**				
Measures of health equity	Moderate interest	Rarely available	No	Little available data and difficult to define.
Days of schooling gained	Low interest	Rarely available	No	Too narrow a measure; not applicable to all pathogens.

* Additional measures suggested by interview respondents. ** Definition of a severe outbreak will depend on pathogen and outbreak setting but may, for example, be outbreaks exceeding certain case thresholds.

**Table 2 diseases-12-00073-t002:** Aligning model design with stakeholder needs at each phase of an investment case.

Framework Phase	Key Decision Points and Considerations
**Problem framing and data sourcing**	First phase of developing a modelling analysis involves considering what questions are being asked, what needs to be modelled to answer them, and who will use the findings [17,18]. There should be a clear understanding of the disease and setting and who intends to use the findings of the study. These factors inform the choice of scenarios to be modelled as they must be sensible for the context and able to answer the planned objectives. Finally, the choice of outputs should meet the needs of stakeholders and be producible by the planned scenarios.
**Model choice**	Once the scenarios and expected outputs have been decided it is necessary to choose or develop a fit-for-purpose model. Must consider the minimum model requirements and complexity to answer the question and how model inputs can be parametrized from the available data. It may become clear that insufficient data are available to inform a model which is complex enough to answer the proposed questions, in which case the scenarios or outputs may need to be reconsidered.
**Model implementation**	The selected model needs to be calibrated to the relevant empirical data for the outbreak with which the analysis is concerned. Calibration, or ‘fitting’ the model to data, is important to ensure model predictions are consistent with reality [33]. There are a range of methods for performing calibration, either manually or via probabilistic techniques, which vary in complexity [22,33]. The specifics are not discussed here as the appropriate method for calibration and assessment of goodness-of-fit will depend on the model chosen and the people performing the analysis. Once a suitable calibration is achieved, the selected model scenarios can be implemented, including the baseline scenario (which would typically be captured by the parameter set produced during calibration).
**Interpretation and communication**	Analysis of results will typically involve the comparison of model outputs between scenarios. Model validation is a critical process here; different types of validation were defined by Eddy et al. and have their own strengths and weaknesses [32]. For analyses which fit within the framework, face, internal, and external validity would be the most feasible and appropriate and should be performed in consultation with stakeholders. Once modellers and stakeholders are confident that a model and results are valid, interpretation and communication can begin. Interpretation involves determining the key messages which can be derived from the results and the extent to which they align with stakeholder expectations. Communication requires clear and concise representation of the key messages and important features of the results, such as uncertainty, as well as the implications of the findings and how they relate to the study objectives and stakeholder needs.

## Data Availability

The model code and result files for the measles and Ebola analyses can be found at <https://doi.org/10.5281/zenodo.10578808>. Data sources recorded during the data search can be found in ‘Review sources.xlsx’. The interview data presented in this study are available upon request from the corresponding author. The data are not publicly available due to ethical requirements.

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
