# Peer review of "A Framework for Assessing the Impact of Outbreak Response Immunization Programs"

_diseases, 2024, doi:10.3390/diseases12040073_

Round 1

Reviewer 1 Report

Comments and Suggestions for Authors

Congratulations. This is a very instructive document about how to meaningfully report about the (potential) benefit of different types of interventions on outbreak response immunization (ORI) back to decisionmakers.

I like to make some comments on the document about making the program more accessible and useful to a broad audience because the approach presented is certainly applicable to many different situations in healthcare assessment than just ORI, I guess. This is therefore not a critic because I really appreciate what the authors have produced, but these are rather some reflections I had when reading the document.

I was wondering in which way the authors were influenced by the many programs we are now seeing promoted on the web of data analyses about Dashboard evaluations of data for organizations (private or public) in the domains of medical and non-medical conditions. Because in a Dashboard assessment the same logic is present as here presented for decisionmakers about what is it that they like to see as critical information about new strategies implemented in an organisation, with the gain by when, how, the sensitivity analysis, and the optimal strategy given the resources available. If it is not the case, one should maybe highlight in the discussion section that what is presented here is not completely out of the blue but finds today its application in many different environments outside the context as defined here and it is maybe useful to make comparisons. One point not presented here but very useful for decisionmakers as done in many Dashboard presentations is about having deployed the best use of resources to optimize outcome, or what is the extra resources needed to reach specific outcome goals by when. These are elements that can easily modelled and calculated within the data and analyses done and that is also often requested by decisionmakers once that info is available. I’m aware of Dashboard constructions in healthcare for assessing Value Based Healthcare in hospital care, for the public health and economic assessment of the infection problem in ageing adults, for the better efficiency working of pharmaceutical plants, to name a few examples.

It is always difficult to select the right information in the main document to precisely clarify the key message of the paper, however there is so much so useful information in the Appendices I’m afraid that this will be lost if the effort is not taken to read it. I would therefore suggest presenting the approach as a special issue of the journal and being able to present the current Appendices into different, separate papers as that may help to highlight the importance of the two cases scrutinized as well as the discussion round with the stakeholders. It is a suggestion, but I guess that it must have been discussed already and I would be in favour of separate papers.

Here are some additional comments:

-I had the impression that this has been developed for LMIC only but it is not mentioned in the title nor in the introduction…

-is it possible that this should be applicable to HICs as well or what makes it that it is less applicable to HICs?

-have you thought to consider Flu as well as COVID outbreaks?

-you go for an assessment about disease control, what about strategies for elimination?

-there is not so much as a surprise as what the stakeholders selected as critical info in their context of LMIC: deaths and cases avoided. What is a surprise, is the DALY concern is not reported as critical information (it is a little in contradiction with what is mentioned in the Appendix A).

-I understand that the colours in Table 1 are defined by the contextual assessment done. In HICs hospitalisation will be a very critical element to consider and should be green coloured.

-the Figures in Model Implementation on pages 12 and 13 should receive some more explanation about what the blue area meant. Are the black dots the reported baseline situations and indicate the blue area where the results of the scenario analysis are moving?

Reviewer 2 Report

Comments and Suggestions for Authors

Here you have my comments. 

INTRODUCTION

Explain what "zero dosis children" means. Note that the reader may not be familiar with the issue.

In the introduction, speak about the Milleniun United Nations' sustainable development objectives, including vaccination. Also, describe the situation of child vaccination at the world level and in Africa.

Material and Methods

More details about the qualitative interviews should be provided: "A series of 16 qualitative interviews were conducted over Zoom throughout November and December 2022 with a total of 25 participants." What kind of interview did you perform? Did you use a focus group?

Did you use any protocol to extract the information from the interviews?

Did you use any software, such as Atlas Ti, to extract and classify the information?

The authors said they used mathematicals models but do not explain how they did it, software are used, etc.

Bibliographic search

In Appendix B, the authors should present information on all the search strategies, and if the boolean operator was used.

Also information about what authors performed the search shoul be provided. Idientifyng them with initials.

In page 13 there are graphics with model implementations, but in the material and methods the authors didnt explain how they computed. Also this graphic should be explained.

Apendix B: Authors said "There was a significant difference in the types of results returned when comparing the different parts of the search", please explain what statistical test were used, the software, and present the data in the text.

Round 2

Reviewer 2 Report

Comments and Suggestions for Authors

The paper can be accepted.